# Citizen scientists and university students monitor noise pollution in cities and protected areas with smartphones

**Lucy Zipf** [ID]*◉, **Richard B. Primack**◉, **Matthew Rothendler**◉

Biology Department, Boston University, Boston, Massachusetts, United States of America

◉ These authors contributed equally to this work.

* lzipf@bu.edu

## Abstract

Noise pollution can cause increased stress, cognitive impairment and illness in humans and decreased fitness and altered behavior in wildlife. Maps of noise pollution are used to visualize the distribution of noise across a landscape. These maps are typically created by taking a relatively small number of sound measurements or simulated on the basis of theoretical models. However, smartphones with inexpensive sound measuring apps can be used to monitor noise and create dense maps of real-world noise measurements. Public concern with noise can make monitoring noise pollution with smartphones an engaging and educational citizen science activity. We demonstrate a method utilizing single-day citizen science noise mapping events and a university lab to collect noise data in urban environments and protected areas. Using this approach, we collected hundreds of noise measurements with participants that we used to create noise maps. We found this method was successful in engaging volunteers and students and producing usable noise data. The described methodology has potential applications for biological research, citizen science engagement, and teaching.

## Introduction

Noise pollution, or unwanted and disturbing sound, is a pervasive feature of modern life [1]. Noise pollution from anthropogenic sources, including road traffic, airplanes, landscaping services, and construction, pose threats to the well-being of humans and wildlife, especially around urban areas, where noise pollution tends to be greatest [2]. Noise pollution is also surprisingly common in protected areas, national parks, and rural areas, where it similarly affects both humans and wildlife [3–7].

The impacts of noise pollution have been well studied in the fields of public health and biology. For mammals, birds, and other animals, noise pollution can interfere with communication, vigilance, and foraging, which in turn can negatively affect fitness and compel individuals to leave a location [8–11]. Human exposure to persistent noise can lead to cognitive impairment, distraction, stress, and altered behavior, as well as reduced enjoyment of the

**Data Availability Statement:** All relevant data are within the manuscript and its Supporting Information files.

**Funding:** This work had no source of funding. LZ and RBP were supported through Boston University and MR volunteered.

**Competing interests:** The authors have declared that no competing interests exist.

outdoors [1]. It can also result in insomnia, high blood pressure, and increased risk of heart attacks [2, 12, 13]. A growing understanding of the dangers associated with noise pollution has produced a compelling need to measure and map noise in a diverse array of settings as well as an opportunity to engage members of the public in the monitoring process.

Over the last 10 years, improvements in smartphone apps and hardware have facilitated their use in increasingly sophisticated and technical citizen science studies. Citizen science and education programs have utilized smartphones to investigate plant phenology, agriculture, air pollution, seismic activity, radiation, acoustics, and to map noise [14–23]. These programs have found smartphones to be valuable as readily-available, accessible science tools to collect high quality data that benefit both researchers and citizen scientists, as well as contribute to educational experiences. For example, the MyShake project demonstrates that privately-owned smartphones can be used by volunteers to record early ground shaking during earth-quakes, potentially allowing for earlier detection and emergency response [19].

For the study of noise pollution, smartphone applications can turn GPS-enabled devices with microphones into accurate sound meters [5, 24–27]. Using a smartphone-based tool, researchers or community organizers can recruit teams of citizen scientists or classes of students to take dense measurements of noise in areas of interest, such as protected areas and urban environments, to produce detailed maps of noise pollution [28, 29]. This technical advance has been recognized by researchers and several projects have utilized crowd-sourced measurements of noise pollution to create noise maps (e.g., Ear-Phone, Noise Tube, Noise Spy, Smart City). These efforts have resulted in the creation of dense noise maps in urban environments in areas including New York City, U.S.A. and Seoul, South Korea [5, 23, 27, 30, 31].

Here we present an approach that focuses on actively engaging citizen scientists in smartphone-based noise monitoring, using the iPhone sound meter app SPLnFFT (Box 1), during single-day events. Single-day sampling efforts are employed widely in citizen science and conservation biology. For example, conservation areas will often host a so-called "bioblitz" to document biodiversity of plants and animals in a given area [32]. These events take place during a single day and have goals including documenting rare taxa, creating an inventory of species present, and introducing the public to local wildlife [33]. Reviews and research projects have shown that these single-day events provide useful data for quantifying individual site characteristics and data from multiple events can be pooled to answer broad research questions [32, 34]. For example, data from single-day citizen science events have been used to document species diversity and distributions [35–37]. Single day events are also accessible to a wide range and ability level of citizen scientists and can engage volunteers who might not be willing or able to participate in longer-term monitoring efforts [38].

In this study, we show that single-day citizen science events can be used to create dense noise maps for urban and conservation areas quickly and at a low cost. The approach presented here differs from that of other efforts like Shim et al. [27] and Ear-Phone [31] in which citizen scientists collected noise measurements using their mobile devices continuously over longer periods of time. Such methods allow researchers to monitor changes in noise over time and to create maps of noise over large areas, but they have limited engagement with citizen scientists and the method of continuous passive monitoring introduces location bias and error associated with phone location (e.g. having the smartphone in or out of a pocket, or in or outside of a building), and may create privacy issues for participants.

Here, we examine the potential for single-day citizen science events to provide a venue for volunteers to learn more about noise pollution and to become custodians of the data they collect and their local environments [40]. Single-day, concentrated monitoring efforts with immediate products provide the opportunity for researchers and participants to work closely and collaboratively throughout the data collection and dissemination process, a practice that

### Box 1. Noise monitoring definitions

SPLnFFT: sound meter application for iPhone; includes location services (GPS), calibration features, and built-in mechanisms for data export via email and Dropbox (Fig 2B)

Decibels (dB): unit used to measure the intensity of a sound; the EPA recommends a maximum 24-hour exposure limit of 55 decibels in residential areas to protect the public from these adverse health effects; this is about the volume of a conversation [39]. A very quiet room is about 30 dBA, freeway traffic is 65–70 dBA, and a loud lawnmower could be up to 80–90 dBA.

dBA: A weighted decibel (dB) scale; emphasizes sound energy at frequencies where humans have their most sensitive hearing thresholds.

L50: median dBA over a sound recording; represents background noise levels and is not heavily skewed by brief, loud noises, such as car horns and bird calls [4]

20 second measurement period: sound recording duration used in this study to capture the background noise of an environment; the app records 5 noise values per second or 100 measurements over 20 seconds; longer measurement periods are less suitable due to the possibility of interference from airplane noise. Volunteers were instructed to avoid taking a 20 second measurement while an airplane was flying overhead.

Instructions for SPLnFFT download, set up, and use are available in the S1 File.

has been shown to help maintain engagement and enthusiasm of volunteers [40, 41]. Through the course of teaching volunteers how and why we monitor noise, monitoring noise, and then presenting them with the resulting noise maps, we aim to show citizen scientists how their efforts can inform positive changes in spaces of concern [42].

In this study, the goal of these single-day noise monitoring events is for researchers and citizen scientists to create maps of noise pollution in areas that contain parks and urban spaces. Maps created with this method can then be used by both the research groups and citizen scientists to further our understanding of the distribution and prevalence of noise pollution in these settings.

## Materials and methods

### Citizen science noise mapping events

To demonstrate the method and test its utility we held three, single-day citizen science noise mapping events in the greater Boston area, Massachusetts, USA during September and October 2017—one each at the Blue Hills Reservation on September 16 (42.2142˚ N, 71.0933˚ W), the Fenway/Longwood area of Boston on September 28 (42.3368˚ N, 71.1012˚ W), and the Hammond Woods on October 7 (42.3228˚ N, 71.1741˚ W) (Fig 1). For each event, we took measurements on a single-day, during a several-hour time period that was outside of rush hour traffic. We chose these locations based on the presence of protected areas, vulnerability to noise pollution, and importance to volunteers.

The Blue Hills Reservation is a 2,400-ha state park located in Milton, Massachusetts, about 24 km south of Boston, and a major regional destination for recreational hiking. It is bisected by a major interstate highway and further divided by busy state roads (Fig 3A). The Fenway/

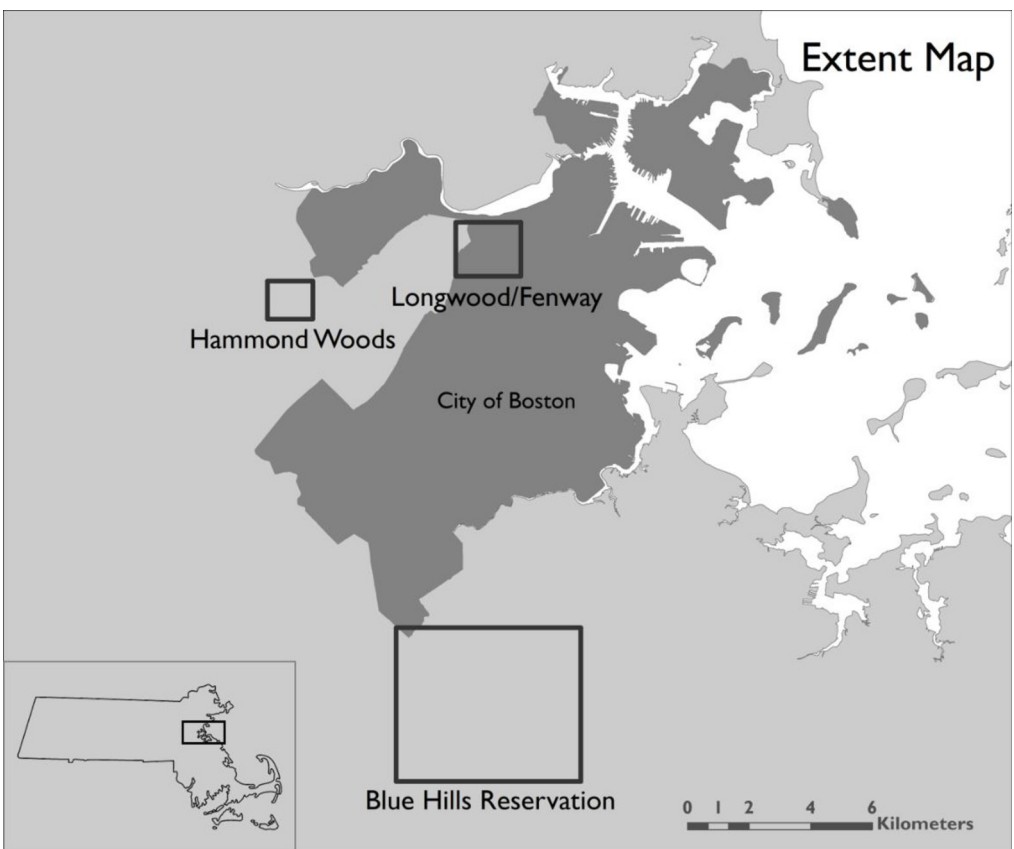

**Fig 1. Locations and sizes of the three noise mapping sites, the Blue Hills Reservation (Milton, MA), the Fenway/Longwood area (Boston, MA), and the Hammond Woods (Newton, MA).** The dark gray region represents the city of Boston, Massachusetts. Source: MassGIS (Bureau of Geographic Information), Commonwealth of Massachusetts EOTSS.

Longwood area is a highly urbanized area of Boston and includes iconic locations, such as Fenway Park (home of the Boston Red Sox baseball team), Museum of Fine Arts, Massachusetts Turnpike, Longwood Medical Area (Harvard Medical School, Brigham and Women's Hospital, and other hospitals and medical facilities), and an urban greenspace known as the Fens (Fig 3B). The Hammond Woods, together with the smaller Webster Woods, is a 46-ha protected area, divided by the Hammond Pond Parkway, within a largely residential area of Newton, Massachusetts, a suburb about 11km west of Boston. This is the largest woods in Newton, and is accessible via two train stops (Fig 3C).

For all three events we partnered with local conservation organizations and university programs—e.g., Friends of Blue Hills, Massachusetts College of Pharmacy and Health Sciences, and the Newton Conservators—to encourage volunteer participation. Participants for the events at the Blue Hills Reservation and Hammond Woods were primarily members of local communities, while most volunteers at the Fenway/Longwood event were graduate students from the Massachusetts College of Pharmacy and Health Sciences (MCPHS) participating as part of a class exercise. Volunteers provided verbal consent to participate in the program at the training sessions. All recruitment monitoring was performed with the safety and well-being of the volunteers in mind; care was exercised when assigning monitoring routes to volunteers to minimize danger to the citizen scientists and the natural environment. Further, the individuals pictured in Fig 2 have provided written informed consent (as outlined in PLOS consent form)

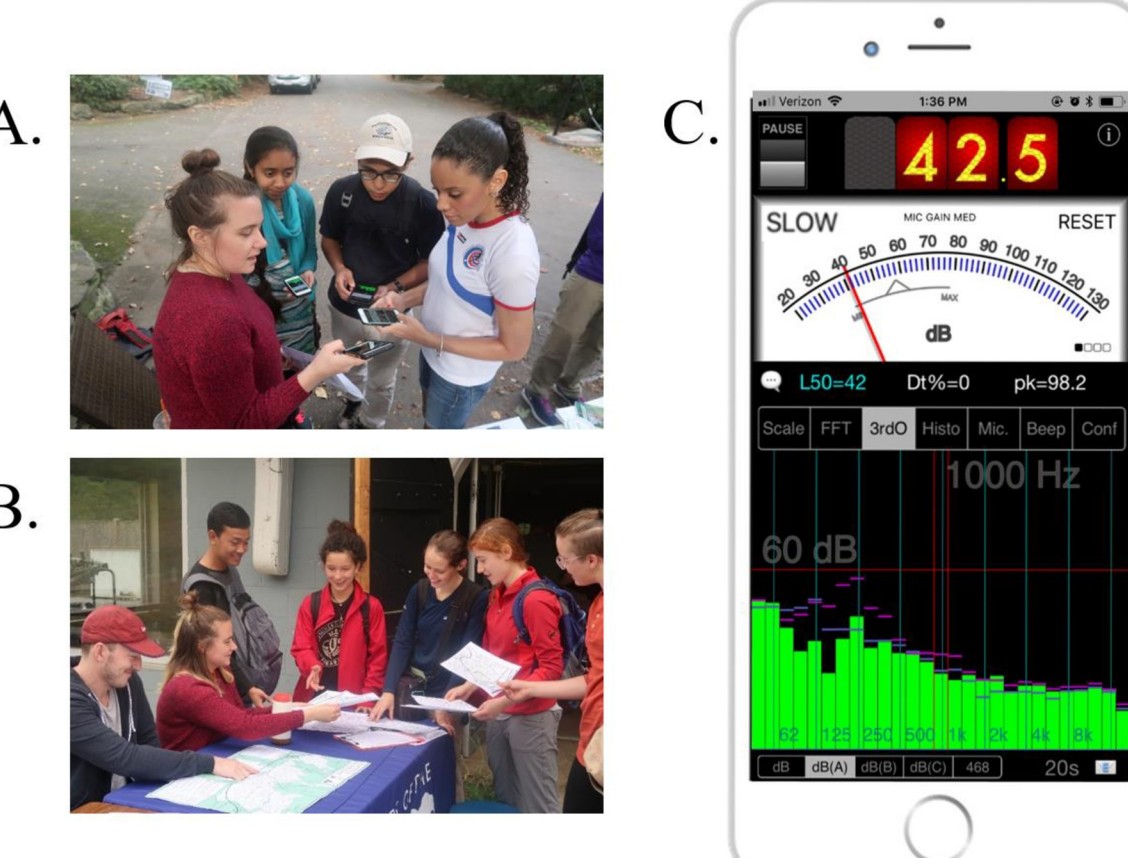

**Fig 2.** (A) Volunteers being trained to record and transmit sound measurements at the Hammond Woods; (B) Volunteers receiving their route assignments at the Blue Hills Reservation, the individuals pictured have provided written informed consent (as outlined in PLOS consent form) to publish their image alongside the manuscript; (C) A screen shot of the SPLnFFT app displaying a sound recording; while the instantaneous noise of 42.5 is prominently display, it is the L50 value of 42, in blue text, that is the primary variable used in our study, source: SPLnFFT by Fabian LEFEBVRE, published with consent from the copyright own.

to publish their image alongside the manuscript. This project does not meet the definition of Human Subjects Research, and therefore does not require IRB oversight.

Researchers, students, and volunteers collected sound data with their personal iPhones using the SPLnFFT app (Box 1, Fig 2). All iPhones are equipped with microelectromechanical systems (MEMS) microphones, though the number of microphones varies across models [25]. MEMS microphones typically capture measurements as low as 30 dB and as high as 120–130 dB, which is sufficient for our method and typical of type 2 sound level meters. iPhones also have GPS and data storage capabilities, features many less expensive sound meters lack. Although using an iPhone-only application may limit the accessibility of the method to citizen scientists, standardizing the microphones among participants is important to reduce variability in the data. Several studies in which researchers map noise with smartphones found that employing a limited number of phone models reduced microphone variability [27, 31]. All iPhones models have very similar hardware and capabilities, whereas Android based smartphones had a wide range of both.

SPLnFFT is well suited to producing noise pollution maps as it has location services, calibration features, instantaneous and time weighted decibel levels, and built-in mechanisms for

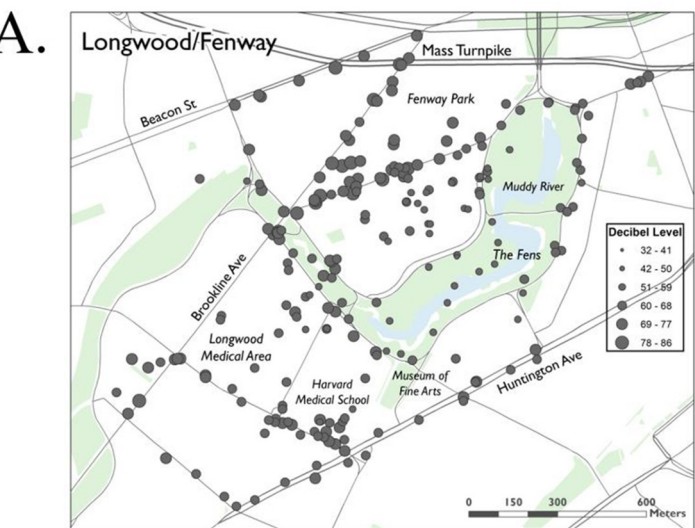

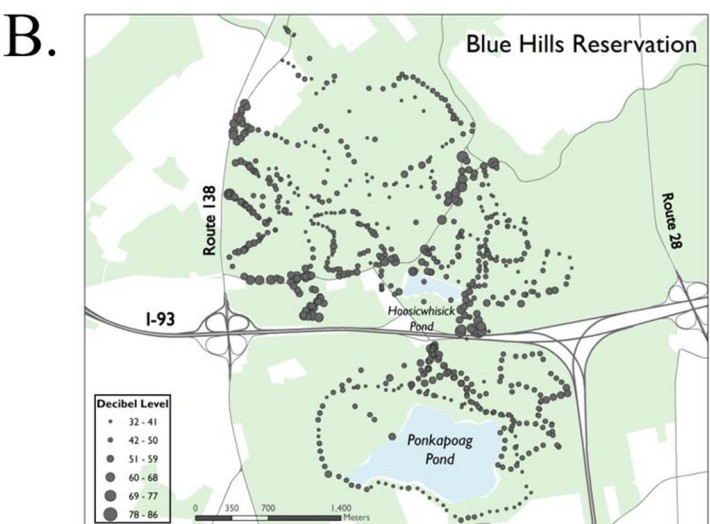

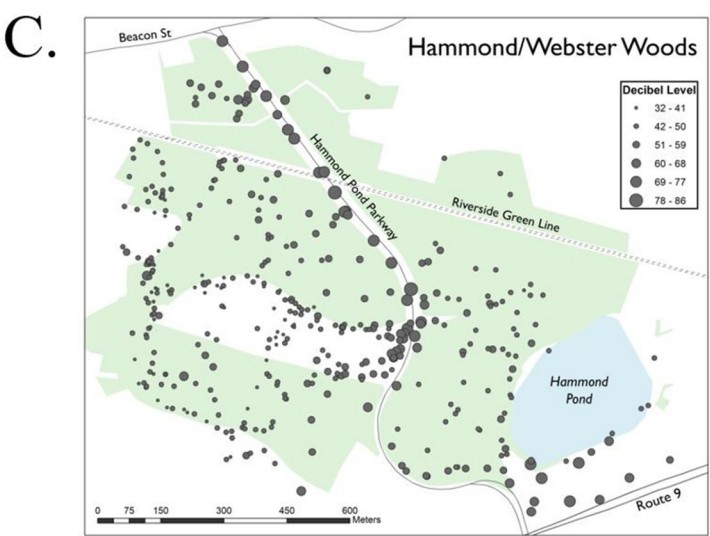

**Fig 3.** Sound maps of the (A) Fenway/Longwood area in Boston, MA, (B) Blue Hills Reservation, and (C) Hammond Woods. Larger dots indicate higher L50 sound values; smaller dots indicate lower L50 sound values. On the maps light blue areas are bodies of water; green areas are parks, and streets and landmarks are labeled. Source: MassGIS (Bureau of Geographic Information), Commonwealth of Massachusetts EOTSS.

data export via email and Dropbox. Recent studies of sound meter applications available for download across iOS and Android platforms found SPLnFFT to accurately and precisely measure noise [25, 26, 43]; specifically, smartphones with the SPLnFFT app were found to record decibel levels with a mean difference of +2.27dBA [25] and +2.74dBA [43] compared to reference levels of a type 1 sound meter. This is consistent with our own testing, which found that an iPhone with the SPLnFFT app recorded decibel levels with a mean difference of +2.77dBA compared to a reference type 2 sound level meter across a range of noises from 35–74 dBA (N = 40, SD = ±0.61 dBA). To achieve type 2 designation, the American National Standards Institute (ANSI) specifies that, among a host of other requirements, error of the recording device must be below ±2.3dB [44]. In addition, Kardous & Shaw [25], Murphy & King [43], and our own testing, found there to be a small standard deviation around mean differences between SPLnFFT and a type 2 sound level meter, indicating high precision (SD = ±0.61 dBA). Overall, SPLnFFT on the iPhone is appropriate for use in our study due to its reasonable precision and the large overall range of noise levels we measure.

In all of our citizen science mapping events, volunteer iPhones were calibrated to a researcher iPhone that had been calibrated using the in-app calibration feature, which aligns sensitivity levels. This calibration to a common standard phone was performed rather than having all volunteers use the in-app calibration feature because the in-app calibration is a lengthy process that requires completely silent conditions, which would be difficult or impossible to achieve during the citizen science events. Additionally, calibrating to a single phone increased the precision across all devices and data points. Calibration of iPhones during an event typically took 5–10 minutes per person. When calibrating iPhones, we had volunteers perform 3–4 test readings using calibrated phones. Phones were considered calibrated when readings were consistently ±1 decibels of the common standard phone, which is minor in comparison to the variation in noise levels (32–85 dB) encountered in this study.

## Volunteer training

Volunteers were encouraged to register for the events in advance, and were then sent handouts via email with a description of the purpose of the project, a guide to downloading and setting up SPLnFFT, and instructions on how to record and submit noise measurements (S1 and S2 Files). Upon arriving at the events, volunteers were warmly greeted, given information (through written materials and discussion) about noise pollution and its effects and mitigation, and assigned specific routes on which they would monitor sound. Researchers then worked with each participant to calibrate their phone to the common standard and demonstrate the noise monitoring methodology. Once trained, the majority of volunteers were then able to collect and send usable data. Our volunteer training approaches were consistent with those of other citizen science studies which also include distribution of training materials prior to the event, on-site training, and skill testing [45–47]. After each event, we improved and clarified our training materials based on volunteer feedback.

Following training, each group of volunteers (1–4 people with a single iPhone) was then asked to follow an assigned transect, stop at semi-regular intervals (~100 m), use the app to record sound for 20 seconds, and submit the L50 median dBA to a repository email address through the app. Routes were established along and near existing trails and roads to ensure the

safety of citizen scientists, and designed to cover as much of the site as possible. Transects were on average about 1–2 km, and took roughly 90–120 minutes for each group of volunteers to complete. They were designed such that volunteers of many ability levels could participate.

Environmental variables, especially wind, can influence noise measurements. The Blue Hills and Hammond Woods events were held on low wind days, but there was some wind during the Fenway event (Table 1). While there are wind shields available for iPhone microphones, we chose to instead instruct the volunteers to shield their phones with their bodies to minimize additional costs and logistical requirements associated with this study. Volunteers were asked to turn their backs to the wind when taking noise measurements and hold the phone about 10 cm from the chest. To test the effects of phone shielding we measured noise both shielding the iPhone with our bodies and facing the phone into the wind on a windy (average wind speed 8.9 m/s, gusts to 13 m/s) day on the Boston University campus in Boston, MA. We found that when there were no gusts of wind, the impact of wind on noise measurements was negligible, averaging only 0.25 dBA lower when facing away from the wind. During periods with wind gusts we found the shielding method to be successful in reducing the influence of wind on the noise reading, with the shielding method resulting in readings that were on average 3.3 dBA lower than when the phone was facing the wind. For the noise maps created using this method we found phone shielding to be sufficient in limiting wind interference.

## Data processing and management

There is no in-app mechanism for storing multiple data points and GPS locations in SPLnFFT. Therefore, each data point, which included the L50, coordinate position, and length of recording, was submitted independently via email by a volunteer as soon as it was recorded. This also allowed researchers to monitor, in real time, data being submitted by each volunteer in the field. If there were abnormalities in the data, or if a volunteer submitted data in the incorrect format, such as forgetting to include the variable of interest (median dBA), it was easy to identify the mistake and call the individual on their smartphone to correct the problem or ask them to report the correct value. GPS errors sometimes, though infrequently, occurred when volunteers lost cellular reception during monitoring; in these cases, SPLnFFT would assign a single location to all measurements taken when reception was lost. These incorrect data were removed from the dataset.

After completing their observations, most volunteers at the Blue Hills and Hammond Woods stopped by the reception table—and the MCPHS students reconvened in the classroom—to discuss their findings and how their data would be processed and mapped.

All data submitted via email were extracted into a Microsoft Excel file using a macro written in Visual Basic for Applications (VBA). Extracted data included location (latitude, longitude), L50 value, sender, and time sent. We created the noise maps using geographic information system (GIS) software.

## Post-event survey of citizen science volunteers

An important outcome of this method is education and engagement with citizen science volunteers. We sought to understand reasons for volunteer participation in order to better host noise monitoring events in the future and meet expectations of current participants [40, 41]. During the week following each citizen science event, we sent each volunteer a follow-up survey via email (S3 File). The voluntary survey was multiple choice with space for volunteers to add comments. The aim of this survey was to understand the ease with which participants

learned to use the app and methodology, their motivation for participating in the event, and what they learned or gained from the experience.

## Results

Across the three citizen science events, we collected a total of 1418 usable observations of noise levels, which we used to generate maps of noise pollution at each location (Fig 3). At each event, we had 16–19 volunteer groups, each consisting of 1–4 people. The number of observations varied from 241 at Fenway/Longwood to 746 at Blue Hills, with the number depending on length of time that observers remained in the field. For each event, about one-seventh of the data were discarded due to user or technical errors (Table 1). The number of points removed from each event were not significantly different from each other ($X^2$, (2, N = 1418) = 0.213, p = 0.89), indicating no difference in error rate between students and community member groups.

### Noise maps

Using the volunteer observations, we created noise maps for the Blue Hills Reservation, Fenway/Longwood, and the Hammond Woods (Fig 3). The mean noise level across the Blue Hills Reservation was a moderate 48 dBA, with values ranging from 32 dBA to 73 dBA (Fig 3A). The Hammond Woods had a similar average noise level of 49 dBA and a range of 37–78 dBA (Fig 3C). The Fenway/Longwood area of Boston, MA was dramatically louder, with an average noise level of 67 dBA and a range of 50 to 85 dBA (Fig 3B).

At all three community events, the main sources of noise were cars and trucks along highways and roads and areas that attract crowds and activity, such as parking lots and streets around Boston's Fenway Park (Fig 3). For example, the primary source of noise in the Hammond Woods was the Hammond Pond Parkway (Fig 3C) with noise levels in the range of 60–78 dBA; the roadway is clearly visible on the map as a band of high noise levels (Fig 3C). In the Blue Hills, areas along the highways and main roads and near the parking lots were the noisiest (Fig 3A). In the Fenway/Longwood area, the loudest noise values were recorded near the Massachusetts Turnpike, but noise levels were also high in the area around Fenway Park and the medical area due to heavy traffic on city roads (Fig 3B). In the two conservation areas the wooded areas far away from roadways were the quietest locations, with values in the Blue Hills Reservation of 32–36 dBA (Fig 3A) and values in the Hammond Woods of 37–50 dBA (Fig 3C). In the Fenway/Longwood area, the quietest areas were in public parks, such as the Fens,

**Table 1. For each citizen science event: Number of volunteer groups, total number of usable observations, the percent of data that had to be removed due to human or instrument error, and average temperature and wind speed on day of event data from the weather station at Blue Hills Observatory in Milton, MA.**

| Area | Blue Hills Reservation | Fenway/Longwood area | Hammond Woods |
|---|---|---|---|
| Number of groups | 19 | 16 | 19 |
| Number of observations | 746 | 241 | 431 |
| Area (km$^2$) | 12 | 1.5 | 1 |
| Spatial Resolution (observations/km$^2$) | 62. | 241 | 287 |
| Percent of observations removed | 14% | 15% | 15% |
| Average temperature | 22˚C | 24˚C | 18˚C |
| Average wind speed | 0.8 m/s | 4.2 m/s | 2.9 m/s |
| Size of mapped area | 3.9 × 3.2 km | 1.6 × 1.3 km | 0.8 × 1.2 km |
| Average L50 (dBA) | 48 | 67 | 49 |
| Standard Deviation L50 (±dBA) | 11 | 7 | 8.5 |

and a nearby residential area with minimal traffic to the northwest (Fig 3B); but even there, the noise levels were 50–65 dBA, which is at least 15 dBA higher than the quiet areas of the Blue Hills Reservation and the Hammond Woods.

## Volunteer feedback

Of the 54 registered participant groups in our three citizen science events, 34 responded to our follow-up survey (57% response rate). Volunteers reported that they participated in the event because they were interested in learning more about noise pollution (25%) or were concerned about noise pollution (17%). The remainder of volunteers participated to help advance conservation science (29%), support a good cause (25%), or for the enjoyment of being outside (4%).

We were also interested in the volunteers' experience with the technical aspects of the method. 65% of volunteers reported learning how to use the SPLnFFT app within 5–10 minutes, with the remainder reporting needing 10 to 20 minutes. 40% of volunteers experienced some problems taking or sending data during the events, mainly due to loss of cellular reception, an important consideration for such citizen science projects. 48% of volunteers stated that they planned to use their smartphones to measure noise themselves in their own environment with the methods we had taught them. As an additional follow up participants were emailed a summary of the noise data collected at the event they participated in, similar to Table 1, and maps of the noise data, similar to Fig 3.

## Discussion

We find that our approach of using smartphones to monitor and map noise pollution was successful in engaging citizen scientists and creating noise maps. At each citizen science event, volunteers were able to gather about three times more data that could be used to create sound maps than the research group could gather alone in the same amount of time.

Generally similar patterns of noise sources and distribution across the landscapes were observed in all three sound maps with noise levels highest near busy roads and parking lots, and lowest in forested areas far from roads. Further, each map showed distinctive spatial patterns of noise pollution unique to that site (Fig 3). Although none of the results presented in this paper will surprise acoustical engineers, hosting single-day citizen science noise monitoring events in areas of research interest allows wildlife biologists, public health researchers, community activists, nature enthusiasts, and students to measure soundscapes with minimal cost and technical expertise. Wildlife biologists, for example, can use this methodology to quickly and inexpensively map noise pollution in a research site, and see how different noise levels impact wildlife behavior [8]. Biologists can also modify our approach by observing changes in wildlife during repeated measurements of noise pollution in an environment where noise levels fluctuate strongly, such as near an airport or train line. Conservation biologists can carry out surveys to determine how noise levels from roads and airplanes flying overhead affect the experience of visitors to nature reserves and other protected areas.

Our survey results also indicated that many volunteers were interested in monitoring noise with our methodology in the future, with 48% of volunteers indicated that they plan to use the smartphone app and methods again. Researchers and members of the public can use the approach (and expand on it) to investigate fine-scale spatial and temporal patterns in noise distribution in their own lives and increase participant understanding about the sources and consequences of noise pollution, as well as methods for its monitoring.

While data collected independently by participants, outside of the single-day monitoring events, might not be appropriate for inclusion in this study or other directed studies of noise in specific areas, the method that we describe in this paper may confer other benefits for the citizen

scientists involved. Volunteers can monitor noise pollution themselves in their own environment rather than relying on the government or academic researchers to provide them with this information, and members of the public can design a survey of noise pollution in a way that fulfills their own needs and concerns [48]. For example, citizen scientists can use this method to record the percent of time that airplane and vehicular noise levels in a conservation area, children's playground, and residential area are over threshold dBA values during time periods of interest.

This sound mapping methodology can also be utilized as a laboratory exercise for high school and college students; a sample lesson plan is available in the S4 File. In the lab exercise, learning outcomes could demonstrate student abilities to (1) understand the concept of noise pollution and tools used to measure noise levels, (2) apply this knowledge of noise monitoring with smartphones to create noise maps around their campus, and (3) evaluate the noise maps and draw conclusions about the local distribution and consequences of noise pollution [49, 50]. These outcomes can be achieved by introducing noise pollution to the class through a 10 to 20-minute lecture on noise pollution, a 15-minute description of the noise monitoring methodology, and a one to two-hour lab exercise in which students take noise measurements of their campus or nearby areas. In the post-lab assignment, students create their own noise maps and describe the patterns of noise pollution that they have encountered. From these maps and observed patterns, students are then able to reach conclusions about noise sources on campus and surrounding areas, potential threats to human and wildlife health, and propose mitigation strategies. This lesson could be customized for students in many disciplines, including public health, conservation biology, ecology, environmental science, and wildlife biology.

In all cases, volunteers in noise monitoring events should have a degree of ownership of the data and conservation objectives laid out by the research group. Though this approach does lend itself to highly controlled studies aimed to answer specific research questions, the purpose of this study is not just to use volunteers to gather data. A significant additional goal of this method is educating citizen scientists about noise pollution and engaging their interest in and concern for the well-being of their community and broader environment [40]. For both citizen scientists and student participants it is important to understand and meet volunteer expectations for learning and the impact of data collection [42]. Here, this was done by presenting background information beforehand, personal training of the methods, providing resulting maps and data to participants, following up with participants about their experience, and maintaining overall communication throughout the project. The maps created by citizen scientists during these single-day mapping events can provide researchers with a starting point for further studies of the effects of noise pollution while also teaching participants a research skill and providing detailed information about their local environment.

## Conclusions

Citizen science noise mapping events as described in this paper are an enjoyable and educational platform from which researchers can engage citizen science volunteers and students in understanding and quantifying noise pollution. Educators and biologists could incorporate this new smartphone noise monitoring technology into their research, outreach, and education programs to gather data. As smartphone technology continues to improve, so too will the accuracy and reliability of smartphone based sound measurements. More work must be done on making noise monitoring apps more user friendly and functional in remote locations.

## Supporting information

**S1 Data.**
(XLSX)

**S1 File. Instructions for noise monitoring application download and set up.** Instructions for citizen scientists to download and set up the SPLnFFT application prior to the noise monitoring event, can also be used as a handout.
(DOCX)

**S2 File. Instructions for citizen science noise monitoring in the field.** Instructions on calibrating the phone, and taking and sending a noise measurement for citizen scientists, can also be used as a handout.
(DOCX)

**S3 File. Post-event survey of citizen science volunteers.** The survey questions that were sent to noise monitoring participants after a noise monitoring event.
(DOCX)

**S4 File. Lesson plan for noise monitoring laboratory experiment for high school or college courses.** A sample lesson plan for a laboratory activity that could be conducted using the described noise monitoring method. The learning outcomes students will achieve in this lab are to (1) understand the concept of noise pollution and tools used to measure noise levels, (2) apply this knowledge of noise monitoring with smartphones to create noise maps around their campus, and (3) evaluate the noise maps and draw conclusions about the local distribution and consequences of noise pollution.
(DOCX)

## Acknowledgments

We greatly appreciate help provided by volunteers from the Newton Conservators and the Friends of Blue Hills, and students from the Massachusetts College of Pharmacy and Health and Boston University. Tim Condon provided extensive help with the noise map creation. Comments on the paper were provided by Rachel Buxton, Kurt Fristrup, Amanda Gallinat, Caitlin McDonough MacKenzie, Dan Mennitt, Abraham J. Miller-Rushing, Jamie Harrison, Nick Ray, and two anonymous reviewers.

## Author Contributions

**Conceptualization:** Lucy Zipf, Richard B. Primack.

**Data curation:** Lucy Zipf, Richard B. Primack, Matthew Rothendler.

**Formal analysis:** Lucy Zipf, Richard B. Primack.

**Investigation:** Lucy Zipf, Richard B. Primack.

**Methodology:** Lucy Zipf, Richard B. Primack, Matthew Rothendler.

**Project administration:** Lucy Zipf.

**Resources:** Richard B. Primack.

**Writing – original draft:** Lucy Zipf, Richard B. Primack, Matthew Rothendler.

**Writing – review & editing:** Lucy Zipf, Richard B. Primack, Matthew Rothendler.

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
