## [Decision Letter · Decision Letter 0]

4 Dec 2019

PONE-D-19-23835

Citizen scientists and university students monitor noise pollution in cities and protected areas with smartphones

PLOS ONE

Dear Ms. Zipf,

Thank you for submitting your manuscript to PLOS ONE. After careful consideration, we feel that it has merit but does not fully meet PLOS ONE’s publication criteria as it currently stands. Therefore, we invite you to submit a revised version of the manuscript that addresses the points raised during the review process.

Please apologize for the delay, due to the late arrival of one of the reviwer reports.

The reviews are very positive and constructive throughout. I decided to follow the "major revision" recommendation of the 2nd reviewer,

for the reasons given in her/his report, However, I also think that the points put forward by the reviewers are quite easy to deal with,

and strongly encourage you to resubmit the paper with the necessary changes.

A contribution of this kind would be most welcome to PLOS ONE from the pov of this editor.

We would appreciate receiving your revised manuscript by Jan 18 2020 11:59PM. To enhance the reproducibility of your results, we recommend that if applicable you deposit your laboratory protocols in protocols.io, where a protocol can be assigned its own identifier (DOI) such that it can be cited independently in the future. For instructions see: http://journals.plos.org/plosone/s/submission-guidelines#loc-laboratory-protocols

We look forward to receiving your revised manuscript.

Kind regards,

Andreas Mueller

Academic Editor

PLOS ONE

Journal Requirements:

1.

2. Please ensure that you include a title page within your main document. You should list all authors and all affiliations as per our author instructions and clearly indicate the corresponding author.

3.

We note that [Figure(s) 1 and 3] in your submission contain [map/satellite] images which may be copyrighted. All PLOS content is published under the Creative Commons Attribution License (CC BY 4.0), which means that the manuscript, images, and Supporting Information files will be freely available online, and any third party is permitted to access, download, copy, distribute, and use these materials in any way, even commercially, with proper attribution. For these reasons, we cannot publish previously copyrighted maps or satellite images created using proprietary data, such as Google software (Google Maps, Street View, and Earth). For more information, see our copyright guidelines: http://journals.plos.org/plosone/s/licenses-and-copyright.

You may seek permission from the original copyright holder of Figure(s) [#] to publish the content specifically under the CC BY 4.0 license. 

If you are unable to obtain permission from the original copyright holder to publish these figures under the CC BY 4.0 license or if the copyright holder’s requirements are incompatible with the CC BY 4.0 license, please either i) remove the figure or ii) supply a replacement figure that complies with the CC BY 4.0 license. Please check copyright information on all replacement figures and update the figure caption with source information. If applicable, please specify in the figure caption text when a figure is similar but not identical to the original image and is therefore for illustrative purposes only.

4.

We note that Figure [xxxx] includes an image of a [patient / participant / in the study]. 

5. Please amend your Ethics statement in the online submission system to clarify the ethics waiver obtained, and outline details of how participants gave consent.

Reviewers' comments:

Reviewer's Responses to Questions

**Comments to the Author**

1. Is the manuscript technically sound, and do the data support the conclusions?

Reviewer #1: Yes

Reviewer #2: Partly

2. Has the statistical analysis been performed appropriately and rigorously? 

Reviewer #1: N/A

Reviewer #2: No

3. Have the authors made all data underlying the findings in their manuscript fully available?

Reviewer #1: Yes

Reviewer #2: Yes

4. Is the manuscript presented in an intelligible fashion and written in standard English?

Reviewer #1: Yes

Reviewer #2: Yes

5. Review Comments to the Author

Reviewer #1: This article reports on a single-day Citizen Science activity in which volunteers use smartphones to measure noise at different locations. Students process the single measurement data into a noise map. The topic of noise, noise pollution and noise protection is of great importance for society, animals and the environment. Occasions like this can help raise the citizens' awareness of noise and provide them with a scientific method to collect and interpret measurement data themselves.

I recommend publishing this article when the following points of criticism can be addressed satisfactorily. Please see the attachment

Reviewer #2: ********************************************************************************************

see attached file

6. PLOS authors have the option to publish the peer review history of their article (what does this mean?). If published, this will include your full peer review and any attached files.

Reviewer #1: No

Reviewer #2: No

---

## [Author Response · Author response to Decision Letter 0]

22 Apr 2020

Major Suggestions from Reviewers 1 & 2:

1. Lacking data and information in the “Materials & Methods” section

There is no information about technical data of the measurements you report, such as sensitivity, accuracy, precision, spatial and temporal resolution

Thanks for this suggestion. We did originally provide a reference for other studies that used the app, SPLnFFT, but we have now provided more technical information, additional reference, and performed our own assessment. See below.

Please add these data, or at least estimates for them. 

• Most importantly:

a) add a comparison measurement with a professional noise measurement equipment. As the data collection with your participant groups has already taken place, a measurement carried out by yourself would be sufficient.

Great point, we added a comparison between our iPhone method and a type 2 sound meter, as well as included more direct references to papers that have assessed the SPLnFFT app. 

b) add a comparison to a theoretical prediction for the decrease of road noise with distance (based on known result for a line/1D source)

We have decided to omit this figure as we did not feel it added to the message of the paper, and instead focus on the method and the resulting noise maps. 

c) Additionally, a discussion of the quality of data obtained in the “Discussion” section would be most useful for you readers.

Through the paper we have included additional information on the quality of noise measurements with the iPhone and the specific methods of our study.

• The suitability to use smartphones in acoustics, also for measuring noise, has already been studied in depth in the physics education literature. It has been shown that they are reliable tools for educational purposes. Since sound measurement with smartphones is one key aspect to enable the CS project in this paper, this should not remain unmentioned.

Good point, we have followed this suggestion and we now discuss these technical points throughout the paper and we cite appropriate literature about the technical capabilities of smartphone noise measurements. We mention this in the methods, and we reference some acoustics examples in the introduction. 

2. It should be better worked out where the added value lies in the recording of one's own measurement data as a citizen. 

• The keyword could be authentic data and ownership of data (see, for example, Tara O'Neill and Angela Calabrese Barton: Uncovering Student Ownership in Science Learning: The Making of a Student Created Mini-Documentary or Enhag and Niedderer: Two dimensions of student ownership of learning during small-group work in physics and the references therein).

Thank you for these additional references; we have drawn from O’Neill as well as other authors to more clearly convey the benefits to the citizen scientists and our intent to connect volunteers to their local environment though the collection of and ownership over real world data with conservation applications. We have included more information in the intro and discussion about the projects benefits for citizen scientists and their ownership of data collection in a local conservation area. References to papers in the education and citizen science literature have also been added.

• A main characteristic of citizen science is that citizens learn to collect and interpret data independently. But in this study you control the data recording very strictly by specifying special calibration rules and even intervene people by telephone if you are skeptical about their measurement data. How is this compatible with the goal of making citizens as autonomous as possible and not degrading them to data producers?

We provide training to the citizen scientists so that they are capable of collecting and interpreting data on their own. Teaching them how to calibrate their phones is important so that their measurements are more precise. The skills that we provide are transferable. Volunteers can monitor noise pollution themselves in their own environment in a way that fulfills their own needs and concerns. 

Additionally, we very rarely had to intervene to correct a data collection mistake – however we see the ability to correct any errors mid-monitoring as an advantage of the method. It is better than allowing a volunteer to spend many hours submitting incorrect and unusable data.

Reviewer 1

1. line 10: Noise distributions are also often simulated simply on the basis of theoretical models. This statement could be added.

Thank you for the suggestion, this has been added.

2. line 14: in in

Thanks! Fixed in line 13

3. line 21 citizen science (2x)

Thanks! Fixed in keywords

4. line 46: Smartphones have also often been used in physics education (research) for learning in

acoustics, e.g., see Hirth, Michael, Jochen Kuhn, and Andreas Müller. "Measurement of sound

velocity made easy using harmonic resonant frequencies with everyday mobile technology." The

Physics Teacher 53.2 (2015): 120-121. And Klein, P., Hirth, M., Gröber, S., Kuhn, J. & Müller,

A. (2014). Classical experiments revisited: smartphones and tablet PCs as experimental tools in

acoustics and optics. Physics Education, 49(4), 412.

In particular, it was emphasized that the accuracy of measurement for educational purposes is

sufficient.

Thank you, we have added these references (line 49-51) and a statement about their success for education in lines 49-51

5. line 49: please provide literature about using smartphones in the context of air pollution, I am

not aware about that

Lim CC, Kim H, Vilcassim MJR, Thurston GD, Gordon T, Chen LC, et al. Mapping urban air quality using mobile sampling with low-cost sensors and machine learning in Seoul, South Korea. Environ Int. 2019; doi:10.1016/j.envint.2019.105022

Good point! The above reference has been added.

6. line 57: create

This has been changed to produce in line 61

7. line 66: so-called bioblitz

Thanks! Added line 71

8. line 77: Please point out that the character of the single-day events what adds the new value

Added information about the value of single day events in lines 68-81

9. line 80-81: It reads as if the combination of data sources would take place in this article. Please attenuate the statement.

Statement amended!

10. Introduction in general: Please give more attention to the work of Shim et al 2016, which was

published in the same journal. They took measurements over the period of one week, including

working with Android devices. Please briefly present this work here.

We have described the work of Shim et al and others more clearly in the introduction including describing their methods and the differences between our approaches (84-91). We have also commented on the models of phones used in ours and other noise studies in the methods (151-160) 

11. Line 91: Here you write about a several-hour time period, elsewhere about a single-day event. Please stay consistent.

Thanks, added single-day event has now been used consistently through the ms.

12. Line 95-104: The mentioned buildings and areas could be highlighted in the pictures, possibly by a label.

Good point, we changed the labeling in the figures so the areas mentioned in text are now labeled on the maps.

13. Line 125-125: I have a few questions about the method: 

Why was the in-app calibration not used?

Thanks - this is now addressed in line 180-190. The in-app calibration requires a quiet space which is not always available in the field. 

How much do the results of an in-app calibration vary compared to a calibration with a standard

instrument? 

In the materials and methods, we added a comparison between our iPhone method and a type 2 sound meter, as well as included more direct references to papers that have assessed the SPLnFFT app. We found the app and iPhone to consistently overestimate noise values by about 2.8 dB in comparison with the type 2 sound meter, which closely corresponds with what others have found. Besides this overestimate, we found the app and iPhone to be very precise across multiple measurements at various decibel levels. In text lines 162-178

Is it still possible to speak of citizen science if such a control is necessary? 

It is standard practice in science to calibrate recording devices before taking measurements. That way, we can increase the accuracy of measurements. In practice, the iPhones were typically within 1 or 2 decibels of each other before calibration, which is minor in comparison to the 40 or so decibels of variation in sound across sites. 

Does the calibration remain constant over the measuring period, i.e. over 90-120 minutes? 

Based on our experience, the calibration remains constant over the 2-3 hour period of people taking measurements. The accuracy does not change over such short time periods. 

As a test we took 10 measurements at 40-60dB and found the average difference between iPhone and type 2 noise meter at time 0 was +2.6dB; when we repeated the same measurements two hours later, the average difference was +2.3dB, which is very consistent. 

What can you say about the measuring range of the app, especially for noises smaller than 30dB? 

MEMS microphones typically capture measurements as low as 30 dB and as high as 120-130 dB, which is sufficient for our method and typical of type 2 sound level meters (in text line 152-153). Our lowest reading was 32 dB in the field, so the lack of sensitivity of these meters to sound levels below 30 should not impact the study. 

Why did the citizens have to send the data after each measurement and why not send the complete data set at the end of the route?

This is addressed in lines (232-242). The main reason is that the app does not readily store the data. Also, this method allowed us to track noise levels and the progress of volunteers in real time. 

14. Line 125: what is the common standard?

This is now more explicitly stated in 180-190. The common standard is a researcher iPhone that we calibrated the volunteer iPhones to.

15. Line 134: Which criteria were used to select the routes and the density of the measuring points along the routes?

This has been added in lines 207-213. The routes were selected along streets, paths, and trails, and the point density was selected to cover the largest amount of area during the single-day events. 

16. Line 160-162: That sounds like a lot of effort. Please comment on the purpose of citizen science. On the one hand you want to have very accurate measurement data and monitor it pretty closely, on the other hand citizens should have the responsibility for the measurement. How does that fit together?

The value of including citizen scientists despite the technical challenges of this approach is now addressed throughout the ms, as well as above under the major concerns. 

17. Line 197: Wind can have a huge influence, as I know from my own measurements. Has it been checked that people have stood with their backs to the wind? Should a windscreen be used for the microphones in the future?

We have added information on tests we ran to determine the effect of shielding the phone with one’s body, and information on the addition of windscreens for the phone in lines 215-119. We found that when there was a modest and steady breeze, there was no detectable effect of using the body as a wind screen; that is, wind did not affect the measurements. Screening the iPhone with one’s body was effective when there was a strong and gusting wind. Using a windscreen and microphone with each iPhone might increase accuracy slightly in windy conditions, but would have added to the expenses and logistical challenges of the volunteers. We provide more detail about this in the paper. 

18. Line 218: What does local regression mean? What is local about it? What equation was used to the data and what is the message from the regression?

We are removing this figure as it does not add to our overall findings and is not discussed in the text.

19. Line 238: Maybe I'm too pedantic there but detailed is a matter of definition. For a particle

physicist, the amount of data you created is very small. Maybe you find a different description.

We now address this issue more directly, and specify that including volunteers yields maps with about 3 times more data than would be possible with the research group alone during the time available. See lines 317-320.

20. Line 244: what exactly is our method. Please explain and summarize it briefly in the discussion

We have now included a brief description in place of the word method in line 317.

21. Line 255: Wouldn't it have been possible to ask the participants afterwards if they had used the smartphone again? The project took place about 2 years ago....

This is a good point, we emailed the volunteers (1/6/20) to see if they had used the app / monitored noise since the event, but received a low response rate (3 total respondents); many of the participants are students and professionals who may have just been returning after the holidays. Therefore we cannot comment on the use of the app following the events. Of three respondents one had used the app to check the noise level within their apartment. 

22. Line 263: You write that volunteers can monitor noise pollution independently. Doesn't that

contradict the high level of control you have tried to achieve?

This discrepancy is more clearly addressed in lines 344-353, as well as above under the major concerns. 

23. Line 294: Are there suitable apps for Android users that you can recommend and that have appeared in the meantime? 

The primary problem with taking measurements across devices is the microphone differences, not the app itself. We chose to use iPhones because 42% of the smartphones owned in the US are iPhones and all iPhones have standard microphones. Android microphones vary across phone types and manufacturers and are therefore much more variable. There are high quality Android monitoring apps, but the microphone variation would likely be too great for this application. We included more information about the limits and advantages of iPhone in 162-178.

Fig. 3 A: Why did you decide to make this division in the legend? It would be

nice to be able to identify the areas mentioned in the text in the maps.

Good point! We changed the maps in Figure 3 to identify the areas in text rather than circle areas on the maps. 

24. Additional References: Please also consider this work: Zappatore, Marco, Antonella Longo, and Mario A. Bochicchio. "Crowd-sensing our smart cities: A platform for noise monitoring and acoustic urban planning." (2017): 53-67.

Thanks! This study has been referenced throughout the intro as we reference past smartphone based noise monitoring programs.

Reviewer 2

1. As you highlight two kinds of participant groups in the title, you should compare them (eg. by a chi2 test) in terms of measurement quality, e.g. % removed (table 1) or other data you might have; for the former it looks as if there is no difference between citizen science participants and college students (of a health science track), and this is interesting for the community.

Great point, we have included a chi2 calculation to compare removed observations between groups.

2. table 1: doesn’t fit on the page

Thanks! We have reformatted table 1.

3. research for and refer to other lit. beyond noise monitoring in the spirit of your approach. eg:

a) earthquakes

Kong, Q., Allen, R. M., Schreier, L., & Kwon, Y. W. (2016). MyShake: A smartphone seismic network for earthquake early warning and

beyond. Science advances, 2(2), e1501055.

b) radioactivity

Laquai, B. Open Geiger citizen science project; Available online: www.opengeiger.de

Bottollier-Depois, J.F.; Allain, E.; Baumont, G.; Berthelot, N.; Clairand, I.; Couvez, C.; Darley, G.; Henry, B.; Jolivet, T.; Laroche, P.; et al.

OPEN RADIATION: A collaborative project for radioactivity measurement in the environment by the public. EPJ Web Conf. 2017, 153,

08002. [CrossRef]

Keller, O., Benoit, M., Müller, A., & Schmeling, S. (2019). Smartphone and Tablet-Based Sensing of Environmental Radioactivity: Mobile

Low-Cost Measurements for Monitoring, Citizen Science, and Educational Purposes. Sensors, 19(19), 4264

Beser, A.M. How Citizen Science Changed the Way Fukushima Radiation is Reported. National Geographic Society Newsroom, 13

February 2016.

Thanks! We have added some of these interesting papers and made more reference to the existing literature on citizen science phone-based monitoring programs in 44-54.

---

## [Decision Letter · Decision Letter 1]

26 Jun 2020

PONE-D-19-23835R1

Citizen scientists and university students monitor noise pollution in cities and protected areas with smartphones

PLOS ONE

Dear Dr. Zipf,

Thank you for submitting your manuscript to PLOS ONE. After careful consideration, we feel that it has merit but does not fully meet PLOS ONE’s publication criteria as it currently stands. Therefore, we invite you to submit a revised version of the manuscript that addresses the points raised during the review process.

We look forward to receiving your revised manuscript.

Kind regards,

Andreas Mueller

Academic Editor

PLOS ONE

Reviewers' comments:

Reviewer's Responses to Questions

**Comments to the Author**

1. If the authors have adequately addressed your comments raised in a previous round of review and you feel that this manuscript is now acceptable for publication, you may indicate that here to bypass the “Comments to the Author” section, enter your conflict of interest statement in the “Confidential to Editor” section, and submit your "Accept" recommendation.

Reviewer #1: All comments have been addressed

Reviewer #2: All comments have been addressed

2. Is the manuscript technically sound, and do the data support the conclusions?

Reviewer #1: Yes

Reviewer #2: Yes

3. Has the statistical analysis been performed appropriately and rigorously? 

Reviewer #1: N/A

Reviewer #2: Yes

4. Have the authors made all data underlying the findings in their manuscript fully available?

Reviewer #1: Yes

Reviewer #2: Yes

5. Is the manuscript presented in an intelligible fashion and written in standard English?

Reviewer #1: Yes

Reviewer #2: Yes

6. Review Comments to the Author

Reviewer #1: The authors did a good job in addressing all issues that I pointed out during the first revision. The paper has improved a lot and I favour publication.

Reviewer #2: all points were addressed, but please correct the following minor issue.

The spatial resolution values in table 1 are given with senseless precision (up to 5 digits and 2 decimals, while you know the area in km2 probably at best with an error of a few %). Please use a reasonable precision throughout the paper.

7. PLOS authors have the option to publish the peer review history of their article (what does this mean?). If published, this will include your full peer review and any attached files.

Reviewer #1: **Yes: **Pascal Klein

Reviewer #2: No

---

## [Author Response · Author response to Decision Letter 1]

1 Jul 2020

Major Suggestions from Reviewers:

1. The spatial resolution values in table 1 are given with senseless precision (up to 5 digits and 2 decimals, while you know the area in km2 probably at best with an error of a few %). Please use a reasonable precision throughout the paper.

Thanks for this suggestion. We fixed the issue and now report the values in whole numbers, which is much more appropriate precision.

---

## [Editor Report · Decision Letter 2]

15 Jul 2020

Citizen scientists and university students monitor noise pollution in cities and protected areas with smartphones

PONE-D-19-23835R2

Dear Dr. Zipf,

We’re pleased to inform you that your manuscript has been judged scientifically suitable for publication and will be formally accepted for publication once it meets all outstanding technical requirements.

Kind regards,

Andreas Mueller

Academic Editor

PLOS ONE
---

## [Editor Report · Acceptance letter]

4 Aug 2020

PONE-D-19-23835R2 

Citizen scientists and university students monitor noise pollution in cities and protected areas with smartphones 

Dear Dr. Zipf:

I'm pleased to inform you that your manuscript has been deemed suitable for publication in PLOS ONE. Congratulations! Your manuscript is now with our production department. 

Kind regards, 

on behalf of

Dr. Andreas Mueller 

Academic Editor

PLOS ONE